# Alice in Suicideland: Exploring the Suicidal Ideation Mechanism through the Sense of Connectedness and Help-Seeking Behaviors

**DOI:** 10.3390/ijerph18073681

**Published:** 2021-04-01

**Authors:** Minh-Hoang Nguyen, Tam-Tri Le, Hong-Kong To Nguyen, Manh-Toan Ho, Huyen T. Thanh Nguyen, Quan-Hoang Vuong

**Affiliations:** 1Centre for Interdisciplinary Social Research, Phenikaa University, Yen Nghia Ward, Ha Dong District, Hanoi 100803, Vietnam; hoang.nguyenminh@phenikaa-uni.edu.vn (M.-H.N.); hongkong.nguyento@phenikaa-uni.edu.vn (H.-K.T.N.); toan.homanh@phenikaa-uni.edu.vn (M.-T.H.); huyen.nguyenthanhthanh@phenikaa-uni.edu.vn (H.T.T.N.); 2A.I. for Social Data Lab (AISDL), Vuong & Associates, Hanoi 100000, Vietnam

**Keywords:** suicidal ideation mechanism, Bayesian inference, sense of connectedness, help-seeking behavior, transcultural psychiatry, mindsponge

## Abstract

On average, one person dies by suicide every 40 s. However, extant studies have largely focused on the risk factors for suicidal behaviors, not so much on the formation of suicidal thoughts. Therefore, we attempt to explain how suicidal thoughts arise and persist inside one’s mind using a multifiltering information mechanism called Mindsponge. Bayesian analysis with Hamiltonian Markov Chain Monte Carlo (MCMC) technique was run on a dataset of multinational students (N = 268) of an international university in Japan. Item 9 in the PHQ-9 was used to survey suicidal ideation. The associations among four main variables, namely, (i) suicidal ideation, (ii) help-seeking willingness (informal and formal sources), (iii) sense of connectedness, and (iv) information inaccessibility (represented by being international students), were tested in four models. Sense of connectedness is negatively associated with suicidal ideation, but its effect becomes less impactful when interacting with international students. The impact of a sense of connectedness on informal help-seeking willingness (toward family members) among international students is also lessened. Informal help-seeking is negatively associated with suicidal ideation, whereas formal help is positive. The findings support our assumption on three fundamental conditions for preventing suicidal thoughts: (i) a high degree of belongingness, (ii) accessibility to help-related information, and (iii) healthy perceived cultural responses towards mental health. Therefore, systematically coordinated programs are necessary to effectively tackle suicidal ideation.

And how do you know that you’re mad?’ ‘To begin with,’ said the Cat, ‘a dog’s not mad. You grant that?’ ‘I suppose so,’ said Alice. ‘Well, then,’ the Cat went on, ‘you see, a dog growls when it’s angry, and wags its tail when it’s pleased. Now I growl when I’m pleased, and wag my tail when I’m angry. Therefore I’m mad.—Lewis Carroll [1]—

## 1. Introduction

The very basic human instinct of self-preservation presupposes that any life is too precious to be ended. However, according to recent data, one person dies due to suicide every 40 s [2]. This means over 800,000 people die by suicide every year, accounting for about 1.4% of all deaths worldwide [2,3]. In Japan, where the suicide rate is one of the highest in the world, the total number of deaths from suicide in October 2020 is higher than the total number of deaths from the new coronavirus disease (COVID-19) up to that point [4]. Suicide is the leading cause of death for Japanese aged 15–39, with students being especially more vulnerable [5]. Particularly, depression, as a major contributor to suicide [3,6], was found to be highly prevalent among international university students in Japan [7,8]. To explain why this is the case, we need to explore how suicidal thoughts arise and persist inside one’s mind, taking into account a complex set of factors.

From the perspective of evolutionary biology, the act of self-killing is seen as maladaptive [9]. It is an act that can only be realized once humans have gained the necessary cognitive skills to contemplate the implications of no longer existing in the world [9]. No single approach, whether Western philosophy [10,11] or Eastern traditions [12,13], portrays suicide as a simple, straightforward subject even as it is commonly understood as some kind of desire for one’s pain, bodily or mental, to stop. A certain cost–benefit judgment likely happens every time one faces suicidal thoughts. Analyses on the components of such a judgment, however, vary depending on the applied theories.

Regarding contributing factors of suicidal ideation, two major schools of thought stand out [14,15,16,17,18,19,20]. Joiner and Van Orden et al. suggested that thwarted belongingness and perceived burdensomeness together lead to dangerous suicidal desire [15,18]. At the same time, O’Connor and Kirtley found the driving factors to be defeat and entrapment [17]. De Beurs et al. later stated that internal entrapment and perceived burdensomeness are the strongest factors. Still, defeat, external entrapment, and thwarted belongingness do not relate as much to suicidal ideation among young adults [14]. However, Gunn et al. and You et al. both shared that social connectedness (or belongingness) is the key factor driving adolescent suicidal thoughts and behaviors [21,22].

The inconsistencies in empirical research suggest a need for a new theory or model that can explain the underlying complexity and dynamics of the suicidal ideation process. Thus, our study aims to demonstrate how suicidal ideation may form and persist using the Mindsponge model—a mechanism of absorbing, analyzing, updating, and ejecting information [23,24]. The mechanism can be used to explain how a sense of connectedness influences suicidal ideation and what puzzle pieces were missing in prior studies that caused the inconsistencies. Moreover, to justify the mechanism, we employed the Bayesian analysis on a dataset of multinational university students in Japan.

More in-depth explanation, validation, and discussion about the mechanism will be presented later in the following sections: Theoretical Foundation, Methods and Materials, Results, Discussion, and Conclusion.

## 2. Theoretical Formulation

### 2.1. Suicide-Related Theories

Currently, two well-known theories have been proposed and advocated to explain the existence of suicidal ideation and engagement in suicidal behaviors. The first theory is the Interpersonal Theory of Suicide (ITS), proposed by Van Orden et al. [18,19,20]. The theory assumes that a suicidal desire is determined by the co-existence of a high level of thwarted connectedness and perceived burdensomeness. The lethal (or near-lethal) suicidal attempts are driven by the capability for suicide. The authors define thwarted connectedness as a feeling of not belonging in the current environment and the perceived burdensomeness as feeling like a burden for others.

The second one is the Integrated Motivational–Volitional Model (IMV) [17]. Its core idea suggests that suicidal behaviors result from a complex interplay of motivational and volitional phase factors. To explain the formation of suicidal thoughts and respective enaction, O’Connor separated the model into three phrases: pre-motivational, motivational, and volitional phrase. The first phase consists of biosocial contexts for suicide (e.g., diathesis, environment, and life events). The second phrase describes how suicidal ideation is formulated through motivational factors (e.g., social problem-solving, coping, memory biases, etc.). Suicidal behaviors only emerge during the third phase after meeting some volitional moderators (e.g., access to means, impulsivity, capability, etc.).

Both theories have generated useful frameworks for researchers to study the formation of suicidal thoughts and behaviors systematically. However, further understanding about the root of suicidal ideation should take into account the underlying mechanism, in addition to the interactions of risk factors from within. As the human mind is a multiplex system, its psychological process should be explained and interpreted using a dynamic model rather than conventional static theories. Moreover, such dynamics and complexity are amplified when putting physical and social influences from the individual’s surrounding environment into consideration. While the ITS and IMV focus on answering the ontological question of what factors influence suicidal ideation and behaviors using the positivism viewpoint, the present study expands to the “how” aspect—particularly, how the influences on suicidal ideation work. The explanation for suicidal ideation’s predictors utilizes the mindsponge mechanism that concerns a person’s cognitive and environmental interactions with better dynamics, generalizability, and explainability.

### 2.2. Mindsponge Mechanism of Suicidal Ideation

#### 2.2.1. Mindsponge Mechanism

In this sub-section, we would like to introduce the Mindsponge mechanism and how it could help explain the formation of suicidal thoughts and behaviors. Every person develops a mindset (or a set of core values) that defines the person’s identity, perceptions, and behaviors. The Mindsponge is a mechanism illustrating how a person can absorb new values and eject waning values conditionally based on contexts. The absorbing and ejecting mechanisms are driven by a multi-filtering information process detecting and connecting insights (or information) among different disciplines as well as using inductive attitudes for plausible reasoning [23,24]. For clarity’s sake, Mindsponge is not only a coping mechanism aiming to solve internal conflicts but rather a broader, more inclusive model of cognition shifting process, which also includes such kind of adaptation.

Information coming from the surrounding environment is judged to whether be kept or dismissed based on its perceived value. The filtering process uses cost–benefit judgment-including both rational and emotional-through many layers. The closer the information moves toward the mindset, the stronger the filtering effect becomes. Information accepted into the mindset is integrated into one’s belief system and will affect subsequent decisions and, thus, may create a loop that accepts and reinforces similar information. Filters are based on core beliefs and contexts simultaneously, which means an individual’s cost–benefit judgments are made based on the inductive reasoning of environmental factors and the mindset’s preferences. Persistently, the mindset absorbs and ejects information so that it can maximize total perceived-benefit and reduce total perceived-cost for an individual in a constantly updating manner.

#### 2.2.2. Process of Suicidal Ideation

Within the scope of this study, we focus on two major questions:How are suicidal thoughts formulated in a student’s mind?How can a student get rid of suicidal thoughts?

First of all, we assume that any students who have suicidal thoughts in mind consider suicide an option when dealing with certain circumstances (e.g., depression, anxiety, perceived burdensomeness, loneliness, etc.). The option of killing themselves is one among many other alternatives, such as seeking help, solving the problems, doing more meaningful activities, etc. Based on the Mindsponge mechanism, an option is judged by the mind. Two conditions need to be fulfilled before an individual considers an “act” to be a viable and preferable option (here, the act is suicide).

The first condition is the existence of suicide-related information within the environment surrounding the individual (or within the approachable range of an individual). The second condition is the filter’s decision to let the information move in closer toward the mindset. For information to get close to the mindset, it needs to pass through the filter’s gate of cost–benefit judgment. If suicide-related information successfully enters the mindset and becomes one of the mind’s core preferences, suicidal ideation happens. Speaking in this sense, an individual acquires suicidal ideation when he/she perceives suicide (or becoming non-existent) as beneficial.

However, what makes an individual feel better off dead? There are multiple reasons, including depression, psychological distress, physical or psychological pain, etc. Thus, we assume that suffering is the perceived cost of living that drives an individual to consider suicide beneficial. An example of clearer demonstration for this cost–benefit judgment is the case of euthanasia-intentionally ending one’s own life to relieve unbearable pain (usually due to untreatable diseases) with help from medical doctors—a practice considered legal in a few countries. However, the perceived cost of living due to suffering will be lessened if an individual can find effective help sources. When such help exists and is accessible, the person’s mind will favor ejecting suicide-related information based on the filter’s updated cost–benefit judgment.

We present here an example to demonstrate the filtering process. The idea of talking to one’s father may meet the following internal inquiries: is he available to talk; do I like or dislike him for the fact that I talk to him; will he be able to help me emotionally or practically; do I even need to talk to someone; etc. Similarly, the idea of jumping from the rooftop will face evaluation: why do I even think about this; will it hurt; is there no other better option; should I seek help instead; is existence worth it, etc. As formerly accepted information moves closer to the core beliefs, its influence on behavior is stronger, but the inquiries will become more fundamental. For example, the decision to rely on family members for support may be judged based on one’s perceived self-worth or even childhood trauma. In contrast, the decision to attempt suicide may be judged based on the instinctive will to live or prevalent religious belief.

So, what is the role of a sense of connectedness in this process? We hypothesize that an individual’s sense of connectedness between themselves and other people acts as a component of the filter—which is driven by the core values within one’s mindset (preferences). Specifically, if an individual’s level of connectedness is high, his/her filter will be less rigorous toward information received from other people and vice versa. Nevertheless, being less rigorous toward the received information does not mean that an individual can absorb help-related information from a priorly perceived source as inaccessible (to certain degrees), which causes a rise in help-seeking behavior’s perceived cost.

Figure 1 is a visualization of the Mindsponge process of suicidal ideation and help-seeking behaviors in two groups of students: international students in Japan and domestic Japanese students. The red nucleus circle represents a student’s core mindset, which stores the core beliefs and values. In the middle, the light blue circle represents the mind’s filter gate, a buffer zone for information to enter and go through the cost–benefit judgment. The outer green circle represents a student’s surrounding social environment. Each inner circle has a white layer. These altogether comprise a three-dimensional filter for inductive attitude.

#### 2.2.3. Research Objectives

Based on the aforementioned theoretical explanation and the higher education context, we speculate that the sense of connectedness is helpful for students to find and absorb useful help-related information, which could eventually lead to a lower suicidal ideation rate. However, if the help-related information is inaccessible, the effect of a sense of connectedness will become weaker in lessening the suicidal ideation rate. To justify this proposition, we devised three steps:

*Step 1*: We examine the effect of a sense of connectedness and its interaction with perceived information inaccessibility on suicidal ideation. Here, we assume that being an international student represents information inaccessibility due to their low language proficiency, limited social network, acculturative purposes, etc. Thus, Model 1 is proposed. Suppose the interaction between the sense of connectedness and being an international student is positively associated with suicidal ideation. In that case, perceived information inaccessibility might associate with a higher probability of suicidal ideation. The inaccessibility to information might increase the suicide’s perceived benefit (e.g., to stop suffering) because it increases the cost of seeking help. To confirm this assumption, we continue to Step 2.
(1)Suicide ~ α+TCC+TCC∗inter_dom

*Step 2*: We examine the effect of a sense of connectedness and its interaction with perceived information inaccessibility on the willingness to seek help from both informal and formal sources. The associations are explored by Models 2 and 3, respectively. If the interaction between the sense of connectedness and being an international student lessens the supposed positive effect of the sense of connectedness toward help-seeking willingness, then information perceived inaccessibility might reduce the willingness to seek help (or raise the perceived cost of seeking help). To test whether help-related information negates suicide-related information, we continue to conduct Step 3.
(2)InformalHelp ~ α+TCC+TCC∗Inter_Dom(3)FormalHelp ~ α+TCC+TCC∗inter_dom

*Step 3*: We examine the effect of willingness to seek help from informal and formal sources on suicidal ideation. Model 4 demonstrates the association. Suppose the willingness to seek help is negatively associated with suicidal ideation. In that case, we could support the notion that help-related information negates suicide-related information and would suggest that the perceived accessibility to help-related information is the missing puzzle of the “sense of connectedness-suicidal ideation” relationship.
(4)Suicide ~ α+ FormalHelp+InformalHelp 

The Mindsponge mechanism that drives the authors to develop the three steps above is illustrated in Figure 1. The authors also want to be transparent about what led to these hypotheses. This study’s two co-authors have spent more than six years in the university selected as the study site and had both gone through periods of depression with suicidal ideation. Through personal experience and observation, they provided some insights and personal experiences for generating the above assumptions. The materials, methods, and results to support these hypotheses are presented subsequently.

## 3. Materials and Methods

### 3.1. Study Site and Participants

This study employed a dataset of students’ mental health and help-seeking behaviors in Japan’s multicultural environment. The data has been employed and validated in several studies [8,25,26]. The data were collected from Ritsumeikan Asia Pacific University (APU)—an international university in Oita Prefecture, Japan, from October to December 2018. At the time of survey collection, there were students from 86 countries and regions and faculty members from 22 countries and regions. The proportion of international students on campus was approximately 50.1%.

Among the total sample of 268 students, 75% were international, and 25% were domestic. Male and female percentages were 37% and 63%, respectively. International students were from various origins, such as East Asia, South Asia, South East Asia, etc., but the majority were from South East Asia regions (45.5%). More details of respondents’ socio-demographic characteristics are presented in Nguyen et al. data descriptor [25].

The survey design and procedure received permission from APU’s Ethical Committee Board Approval Number 2018-03 and strictly conformed to the World Medical Association (WMA) Declaration of Helsinki. The approval letter is included in the Appendix A. The survey was conducted using the Google Forms platform due to its advantage of accessibility, data management, and confidentiality. The questionnaire was available in both English and Japanese language. Before participants fill in the questionnaire, a consent form with clear explanations of the survey’s purpose, content, information security, and the right to refuse or withdraw was given on the first page and accompanied by in-person presentations.

### 3.2. Statistical Analysis and Variables

While the dataset covers multiple variables, we purposely selected variables that help explain the formulation mechanism of suicidal thoughts among international and domestic students. In total, we employed five variables. The description of each variable is presented in Table 1.

In this study, we employed item 9 in the PHQ-9 to examine suicidal ideation in students. The item was found to be a robust predictor of suicide attempts and deaths regardless of age [30]. The item is “Over the last two weeks, how often have you been bothered by thoughts that you would be better off dead or of hurting yourself in some way?” with four response options: "not at all", "several days", "more than half the days", and "nearly every day". Respondents answering "not at all" are coded as 0, while the rest are coded as 1. For *InformalHelp*, we purposely selected the willingness to seek help from parents and relatives because they are the two most apparent help-seeking sources that international students perceive as relatively more inaccessible than domestic students. For *TCC*, we purposely exclude items of affiliation to keep consistency with the variable *InformalHelp* and to focus on the sense of connectedness.

All the data were analyzed using the Bayesian analysis with the Hamiltonian MCMC technique (Markov Chain Monte Carlo). We employed Bayesian modeling for three major reasons. Firstly, evidence in the current study is novel, which requires future validation in various contexts and benefits from the Bayesian inference’s updating capability [31,32]. Secondly, for justifying the Theoretical Foundation section’s assumptions, high flexibility in model fitting is required. Such flexibility can be achieved by using the Bayesian inference [33]. Finally, given the dataset’s small sample size, the estimation’s precision will be greatly enhanced by applying the MCMC simulation technique [34].

For conducting the analysis, we initially downloaded the data from its online repository and then generated necessary variables (*InformalHelp* and *FormalHelp*) in addition to the existing ones. It should be noted that the variable *TCC* is not available in the published dataset, so we have attached a sufficient dataset and codes in the Appendix A for future replication. The data were later uploaded to the R software (version 4.0.2) and analyzed using the **bayesvl** package [35,36,37]. The package has been utilized for multiple studies in various fields [38,39,40,41]. The simulation setup was organized with 5000 iterations (2000 are warm-up iterations), four Markov chains, and four cores. We used the Pareto smoothed importance-sampling leave-one-out cross-validation (PSIS-LOO) approach to check for goodness-of-fit [42]. The Gelman and autocorrelation plots were also visualized to check whether the Markov property is achieved. Since no study of the same type has been conducted, all estimations’ priors were set as uninformative [31], which is equivalent to a normal distribution with mean = 0 and standard deviation = 10.

## 4. Results

To test the assumptions made above, we examined four models as presented in the Research Objectives sub-section. All the models’ simulated results and their technical validity are presented accordingly.

### 4.1. Model 1

In the first model, we examine the association of a sense of connectedness and its interaction with perceived information inaccessibility (represented by being international students) against suicidal ideation. The logical network of the Model is presented in Figure 2.

Good Pareto k estimates (k < 0.5) in the PSIS diagnostic plot show that the model fits the data (see Figure 3). As shown in Table 2, all simulated posteriors’ effective sample size (n_eff) is larger than 1000. Each trace plot with four Markov chains fluctuating around a central equilibrium shows good convergence (see Figure 4). There is almost no difference between the between-chain variance and within-chain variance as the shrink factors reduce to one during the warm-up phase (see Figure A1). The Markov property is held because the autocorrelations of MCMC-simulated samples eliminated quickly to 0 (see Figure A2).

Sense of connectedness shows a negative association with suicidal ideation ($\mu_{TCC\_Suicide}$ = −0.93 and $\sigma_{TCC\_Suicide}$ = 0.07), but when the interaction with being international students is added, the sense of connectedness’s effect on suicidal ideation becomes less impactful ($\mu_{TCC*Inter\_Dom\_Suicide}$ = 0.07 and $\sigma_{TCC*Inter\_Dom\_Suicide}$ = 0.09). Visually, the interaction effect on suicidal ideation is moderately reliable because the majority of the posterior distribution lies on the positive side (see Figure 5).

### 4.2. Model 2

The second model examines the effect of a sense of connectedness and its interaction with perceived information inaccessibility (represented by being an international student) on informal help-seeking behavior. Its logical connection is shown in Figure 6.

Similar to model 1, the second model exhibits a high goodness-of-fit with the data (k < 0.5, see Figure 7). The effective sample size (n_eff > 1000) and Gelman shrink factor (Rhat = 1) of all simulated posteriors portray a good convergence of Markov chains (see Table 3). Posterior coefficients’ trace plots, Gelman plots, and autocorrelation plots all demonstrate a good convergence signal (see Figure 8, Figure A3, Figure A4).

The total sense of connectedness is found to positively influence informal help-seeking behaviors ($\mu_{TCC\_InformalHelp}$ = 0.31 and $\sigma_{TCC\_InformalHelp}$ = 0.10), which is relatively intuitive. In contrast, the interaction produces a negative value, suggesting the lower effect of sense of connectedness on help-seeking behavior among international students ($\mu_{TCC*Inter\_Dom\_InformalHelp}$ = −0.09 and $\sigma_{TCC*Inter\_Dom\_InformalHelp}$ = 0.04). The distribution of *TCC*Inter_Dom* falls almost entirely in the 95% the Highest Posterior Distribution Intervals (HPDI) range, indicating high reliability of the effect (see Figure 9). We can also see that the majority of simulated samples are located on the positive side of *TCC* (*x*-axis) and the negative side of *TCC*Inter_Dom* (*y*-axis) in the pairwise distribution graph (see Figure 10). These distributions, again, hint at the high reliability of *TCC*’s and *TCC*Inter_Dom*’s impacts on *InformalHelp*.

### 4.3. Model 3

The third model has a similar design as model 2, but the outcome variable *InformalHelp* was replaced by *FormalHelp*. The network in Figure 11 demonstrates the logical connection of the third model.

As all the Pareto k estimates are below 0.2 (see Figure 12), it is plausible to say Model 3 fits the data well. Other diagnostic graphs also show good statistical validity for the simulation (see Figure 13, Figure A5, Figure A6).

Interestingly, the level of sense of connectedness has a negative impact on the willingness to seek help from doctors and professionals. Moreover, when the factor of being an international student is considered, the sense of connectedness’s effect on formal help-seeking willingness is strengthened (see Table 4). Both effects are moderately reliable, as posterior distributions lie primarily on the negative side (see Figure 14).

### 4.4. Model 4

The logical network of Model 4 is illustrated in Figure 15. Using this model, we examine the effect of willingness to seek help from informal and formal sources on suicidal ideation. The PSIS diagnostic plot suggests that the model fits the data well (see Figure 16).

As shown in Table 5, all the simulated posteriors’ effective sample size (n_eff) is larger than 1000, and the Rhat value equals 1. The presented statistics indicate good convergence of the Markov chains. The Markov chains’ good convergence is also confirmed visually by the trace plots, Gelman plots, and autocorrelation plots (see Figure 17, Figure A7, Figure A8).

Figure 18 shows the posterior distributions of *FormalHelp* and *InformalHelp* coefficients against the outcome *Suicide*. The distribution of *InformalHelp* lies entirely on the negative side, hinting at the robustly negative association of *InformalHelp* with *Suicide* ($\mu_{InformalHelp\_Suicide}$ = −0.51 and $\sigma_{InformalHelp\_Suicide}$ = 0.10). In contrast to *InformalHelp*, the *FormalHelp*’s mean is positive ($\mu_{FormalHelp\_Suicide}$ = 0.09 and $\sigma_{FormalHelp\_Suicide}$ = 0.10). Even though the standard deviation of *FormalHelp* is larger than the mean, the majority of simulated samples are located on the positive side, suggesting a moderate-confident positive association between *FormalHelp* and *Suicide* (see Figure 19).

## 5. Discussion

The current study is one of the first to demonstrate the suicidal ideation process of an individual using a dynamic and complex mechanism—the Mindsponge mechanism. We employed the Bayesian statistics with the Hamiltonian MCMC technique on a multinational dataset for justifying the underlying mechanism of suicidal ideation among students. 

### 5.1. Sense of Connectedness: A Gatekeeper

Our results show that sense of connectedness is positively associated with the willingness to seek help from informal sources (parents and family members) and negatively associated with suicidal ideation. Moreover, the negative association between the willingness to seek informal help (parents and relatives) and suicidal ideation is also observed. Thus, it is plausible to say that a high degree of connectedness to other people helps lessen the probability of suicidal ideation by increasing the willingness to seek help from parents and family members. Parents and relatives are normally a good source of helpful and supportive information, especially against one’s death or suicidal thoughts. The results are consistent with past findings that suicidal thoughts and behaviors are negatively influenced by objective isolation, subjective loneliness, and thwarted belongingness among adolescents [20,21,22,43].

Given the fact that a student’s cost–benefit judgment of help- and suicide-related information is deemed uncertain at first and needs deeper evaluation to decide on acceptance or rejection, a high sense of connectedness provides a “priority-pass” (increased benefit perception) for information from an individual’s social network—in this case, parents and family members. In other words, a stronger sense of being connected to parents and family members improves the student’s trust in the information provided by them, including the help-related one. Subsequently, the availability of help-related information increases the cost of suicidal ideation and makes suicide-related information easier to be rejected from the mindset. Nevertheless, things are not always like what we expect. Sense of connectedness might be a good-information gatekeeper providing a “priority pass” to help-related information, but what if the help is not available or perceived to be inaccessible?

### 5.2. Perceived Information Inaccessibility: The Elusive Helping Hands

The positive impact of a sense of connectedness on suicidal ideation does not necessarily mean it is a sufficient factor for preventing suicidal ideation. Our findings support the assumption that a high sense of connectedness is a condition that needs to be satisfied for suicidal ideation prevention. We find that the sense of connectedness’s interaction with being an international student lessens the effect of sense of connectedness on both suicidal ideation and informal help-seeking willingness. In other words, even though international students acquire the merits of a high sense of connectedness level, the merits are seemingly lower than domestic students since the help from parents and relatives is perceived as more difficult to access or unavailable. Thus, the perceived accessibility of help-related information is also a condition that needs to be satisfied for negating suicidal ideation in a student’s mindset.

In detail, the informal help-related information available for an international student is much less effective than that for a domestic student because family members of international students are not physically available nearby. This factor can significantly increase the perceived help-seeking cost in the cost–benefit judgment when a student faces emotional difficulty. When the perceived value of the information decreases due to external barriers (e.g., geographical distance), the mind may hastily evaluate it as inaccessible and reject the information regardless of internal preferences and traits, such as a sense of connectedness. Furthermore, distance barriers limit the amount and intensity of help-related information coming to the mind, reducing its overall presence concerning suicide-related information.

These results suggest that the construct of connectedness is not a fundamental predictor of suicidal ideation but rather a psychological trait affecting the process of filtering suicide-related and informal help-related information. A willingness to connect to and feel belonged to the wrong social groups can lead to negative outcomes [44]. Students in distress who lack valuable sources of help-related information can be attracted to depressive peers or suicidal-oriented online media for temporary emotional sympathy or for whatever personal reasons. However, the harmful influence can be literally deadly, defined as suicide contagion [45,46]. In fact, group suicides are present among adolescents and performed in many different forms within Japanese society [47].

### 5.3. Collective Perception Issues: The Painted Garden

We also suggest another condition that may need to be satisfied for effectively preventing suicidal ideation besides the sense of connectedness and perceived help-related information accessibility. This third factor is the perceived cultural responses towards mental health problems. Our result shows a positive association between formal help-seeking willingness and suicidal ideation, which is likely the result of complex interactions rather than simple factors. We present our explanation below but also acknowledge the possibility of other fitting interpretations. This requires further examination in future studies.

Students with a higher likelihood of having suicidal ideation tend to have a higher willingness to seek doctors and professionals for help, which is quite plausible given the Japanese context and the sample’s student origins (mostly from countries in South-Eastern Asia and Eastern Asia). Professional mental healthcare is relatively less common in Asian countries than in the West because of myriad underlying reasons [48]. However, the most significant cause might be the high social stigma against mental health issues and lack of investment and legislative support for mental healthcare [49,50]. The stigmatization results in the social disapproval, discrimination, and devaluation of families with mentally ill people, causing the student’s hesitation to seek professional help when facing mental health issues. In other words, negative perceptions towards mental illness as a social norm considerably raise the students’ perceived cost of seeking help from doctors and professionals. This speculation is supported by one of our findings that students with a relatively higher level of connectedness may be less willing to seek help from doctors and professionals, although this interpretation requires further examination to be justified.

Suppose the hesitation to seek formal help persists. In that case, students’ mental problems (e.g., depressive disorder, anxiety, stress, etc.) might gradually accumulate and develop to a certain higher level. Hence, they start to perceive suicide as a “promising” option [51,52]. Concurrently, the cost of losing their own life also push students to recall more acceptable options, that is, to seek professional help.

The perceived cost of seeking professional healthcare is amplified for international students. We find that the sense of connectedness’s interaction with being international students lessens help-seeking level. This result again highlights the importance of help-related information accessibility and cultural perception of mental health issues. Specifically, international students have less access to formal help-related information than domestic students due to their low language proficiency and unfamiliarity with the new environment [53,54]. In addition, a local negative collective perception about mental health issues might also hinder international students’ access to formal help-related information due to the fear of social stigmatization, disapproval, and discrimination within the new community.

Based on the above findings, we advocate a more systematic and dynamic approach in studying mental health problems and psychiatric issues, like the information absorbing and ejecting process of the Mindsponge mechanism. Further, tackling the issue of suicidal ideation among students or even among other populations is not only the work of any single intervention but a set of systematically coordinated programs. These programs have to improve the individual’s sense of belongingness (e.g., through social and recreative activities, etc.), increase the accessibility of help-seeking sources (e.g., through mental healthcare services, etc.), and reduce improper perceived cultural responses (e.g., through shifting cultural perception, etc.). 

### 5.4. Limitations and Research Agendas

The study is not without limitations [55]. First, the study employs cross-sectional data for justifying assumptions on a dynamic mechanism. Even though it is still plausible, longitudinal studies testing the reliability of these results and assumptions are necessary. Second, we acknowledge that the effect sizes are small. However, given the study is a preliminary attempt toward a new approach, we believe the consistency in all four models can justify the significance of interpretation or the direction for future endeavors. In addition, in psychological research, effect sizes need to be interpreted carefully to avoid mistakes [56]. Third, for the same reason (initial study), we used uninformative prior to the analysis. However, after testing with prior tweaking, we found the same results, which justified the robustness of the models. Fourth, we acknowledge the small size of the sample. However, regarding the properties of psychological research, we did try to optimize accuracy through our chosen methodology, Bayesian analysis [57].

Last but not least, the Mindsponge mechanism is a very dynamic and complex model; its utility has not been fully employed in the current study. Thus, some multifaceted aspects and problems could not be addressed. Such aspects and problems might be categorized into three groups:The mindset’s values: socio-cultural values, political viewpoints, traits, experience, etc.The filter: the sense of belongingness, habits, inductive attitude, etc.Information accessibility: school environment, healthcare service availability, interpersonal network, etc.

Considering all the aforementioned limitations, the current paper is intentionally presented in the greatest detail and backed up with open resources (data and code) for improving scientific transparency [55,58]. During the current reproducibility crisis, such transparency can help smoothen and lessen the cost of future replications of the study’s results and ways of conducting psychological research [59,60,61,62].

## 6. Conclusions

In this study, we use the Mindsponge mechanism—a mechanism of absorbing, analyzing, updating, and ejecting information as the theoretical basis to present that suicidal ideation is a process of receiving, evaluating, and integrating suicide-related information—which is driven by the individual’s inductive cost–benefit judgments. Within this process, the individual’s sense of connectedness is necessary for seeking help that negates suicide-related information. However, connectedness alone is not sufficient, but information accessibility and the perceptions about the help-related information are also crucial.


**Alice in Suicideland**

*Alice followed a white rabbit into a strange world. There were many guests coming to visit the rabbit in his house. While the Shadows sent by the Queen of Abyss persuaded the rabbit into surrender, other animals tried to offer him help to fight back. A friendly Gryphon was standing at the house’s gate, welcoming nice guests while blocking the vile Shadows. Alice stood and observed all kinds of interesting characters and events in the noisy house. The rabbit kept trying to call his parents at home for supply, but every time he reminded himself that they were staying in the Farland and decided to hang up before even saying a word. There was a wise caterpillar wearing a doctor outfit trying to come in, but his coat was painted with exotic patterns that scared the rabbit. There was a mysterious cat with angelic fur and an evil grin lying on the balcony, with only his shiny fur visible when nice guests are in the house and only his dark teeth visible when the Shadows are inside. There was also a friendly turtle that looked so sad that he made the rabbit cry as well. However, as more and more merry animals got inside the house, the once confused rabbit became determined to reject all coming Shadows. Everyone held a grand tea party and invited Alice, but just then, she woke up. What a weird yet somehow useful dream!*
--------Characters borrowed from Alice in Wonderland--------

## Figures and Tables

**Figure 1 ijerph-18-03681-f001:**
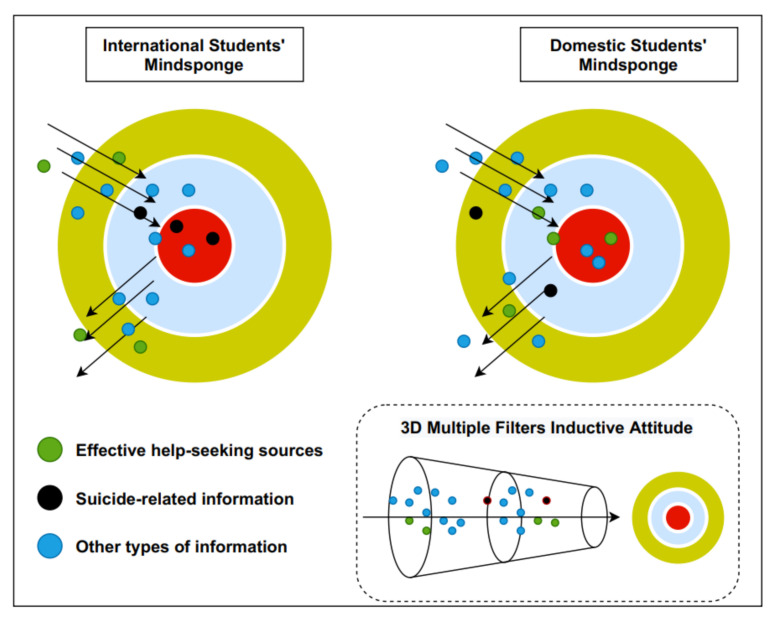
The Mindsponge process of suicidal ideation and help-seeking information.

**Figure 2 ijerph-18-03681-f002:**
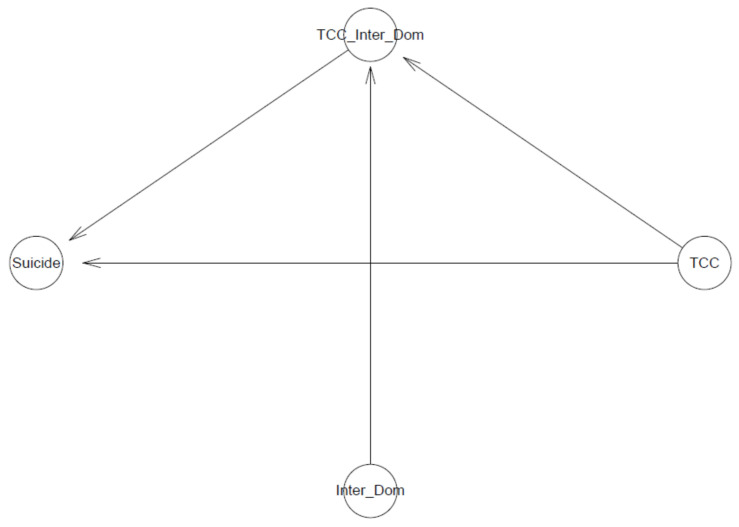
Model 1′s logical network.

**Figure 3 ijerph-18-03681-f003:**
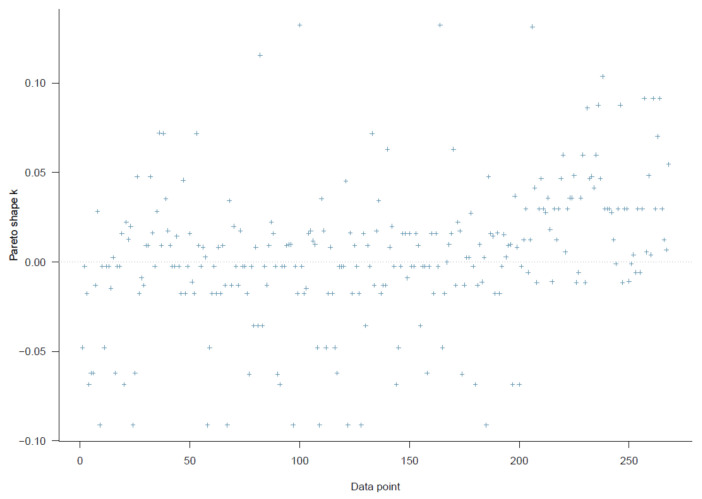
Model 1′s Pareto smoothed importance-sampling (PSIS) diagnostic plot.

**Figure 4 ijerph-18-03681-f004:**
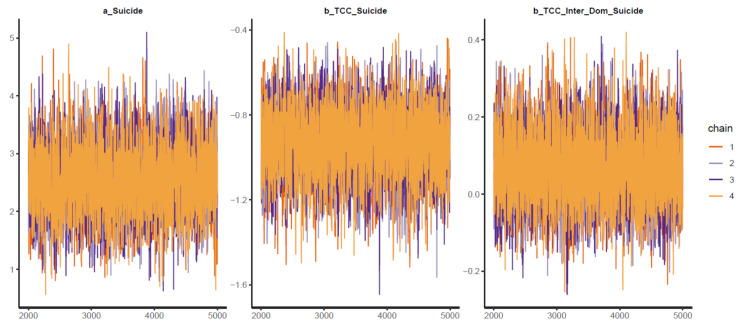
Trace plots for Model 1′s posterior coefficients.

**Figure 5 ijerph-18-03681-f005:**
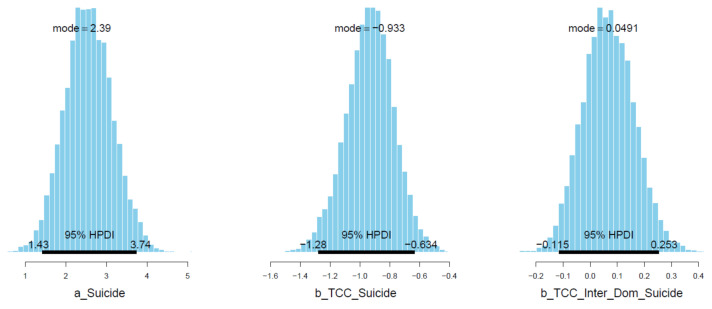
Distributions of Model 1′s posterior coefficients with HPDI at 95%.

**Figure 6 ijerph-18-03681-f006:**
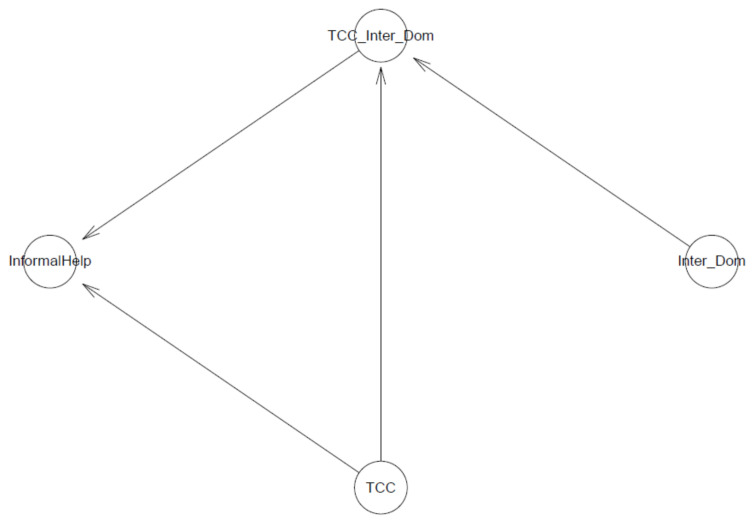
Model 2′s logical network.

**Figure 7 ijerph-18-03681-f007:**
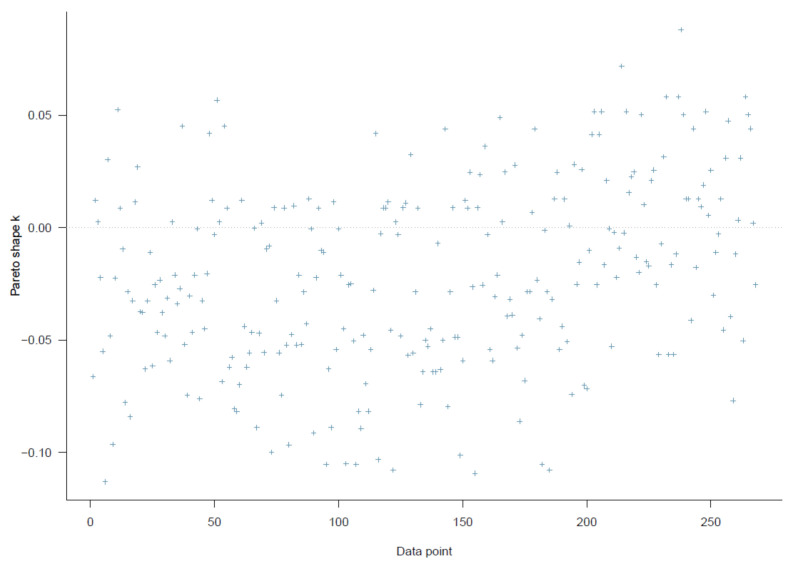
Model 2′s PSIS diagnostic plot.

**Figure 8 ijerph-18-03681-f008:**
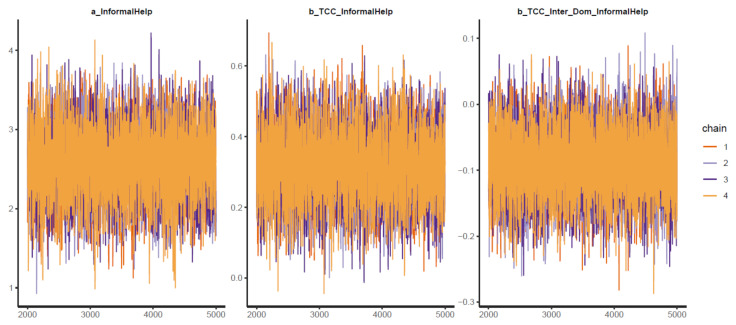
Trace plots for Model 2′s posterior coefficients.

**Figure 9 ijerph-18-03681-f009:**
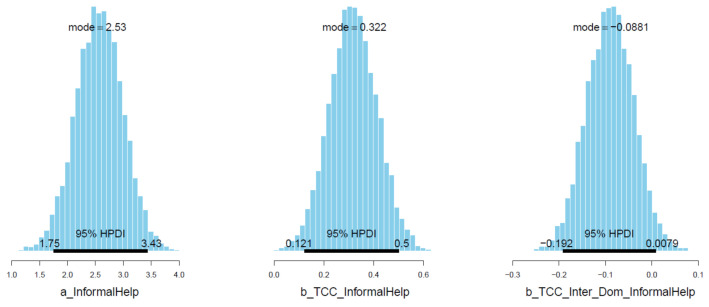
Distributions of Model 2′s posterior coefficients with HPDI at 95%.

**Figure 10 ijerph-18-03681-f010:**
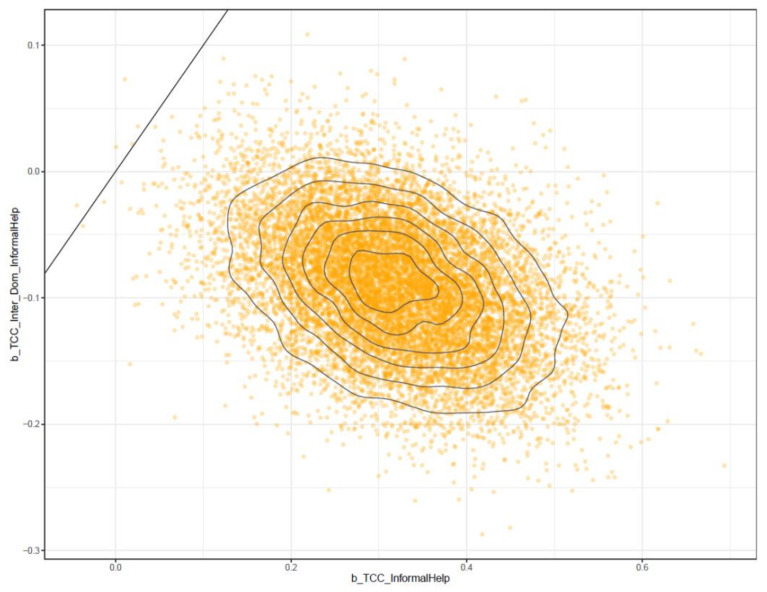
Pairwise distribution plot for model 2′s *TCC* and *TCC*Inter_Dom*.

**Figure 11 ijerph-18-03681-f011:**
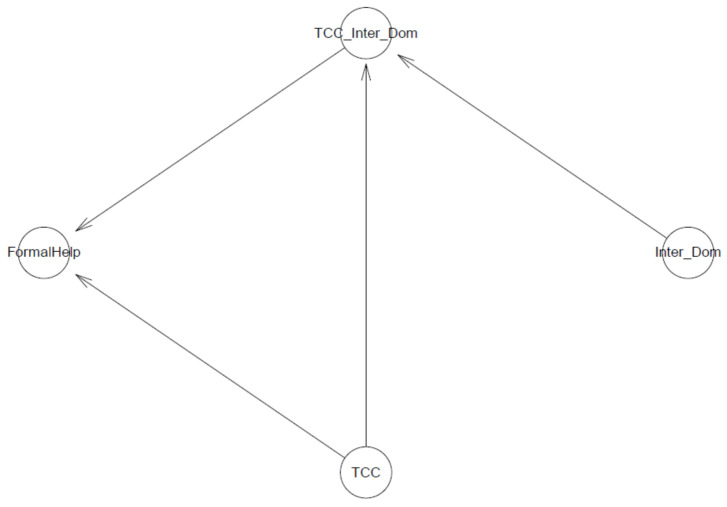
Model 3′s logical network.

**Figure 12 ijerph-18-03681-f012:**
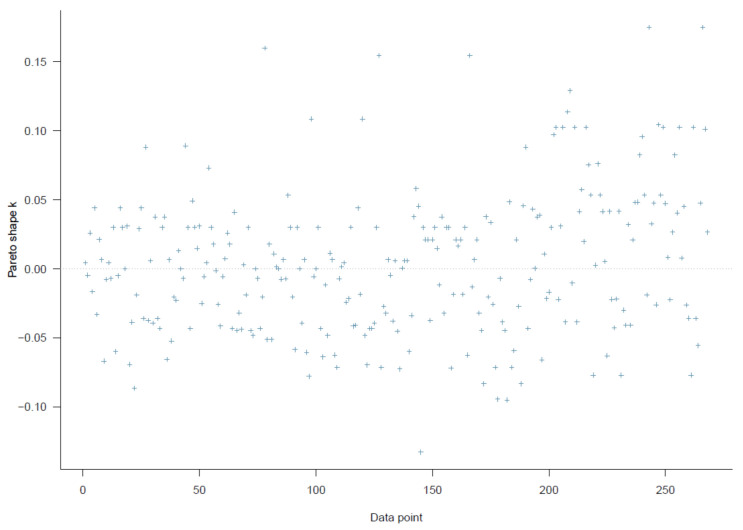
Model 3′s PSIS diagnostic plot.

**Figure 13 ijerph-18-03681-f013:**
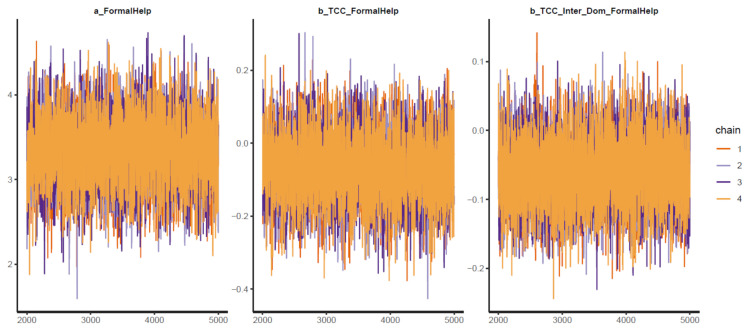
Trace plots for Model 3′s posterior coefficients.

**Figure 14 ijerph-18-03681-f014:**
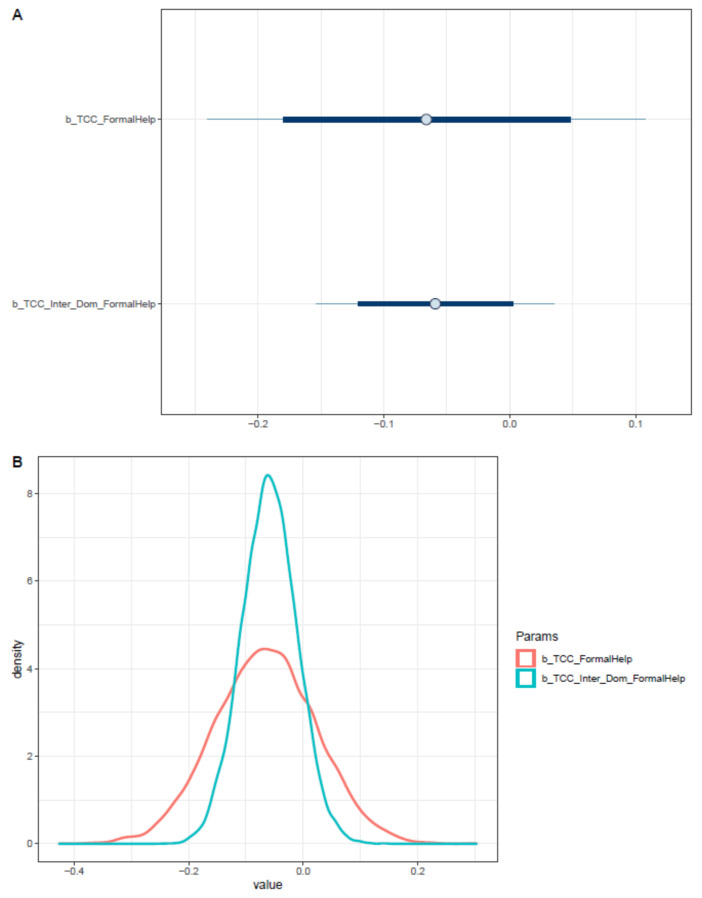
Distributions of Model 3′s posterior coefficients. (**A**)—Interval plot; (**B**)—Density plot.

**Figure 15 ijerph-18-03681-f015:**
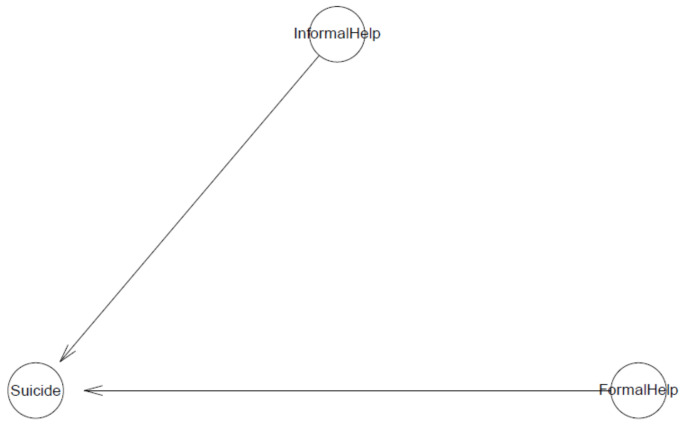
Model 4′s logical network.

**Figure 16 ijerph-18-03681-f016:**
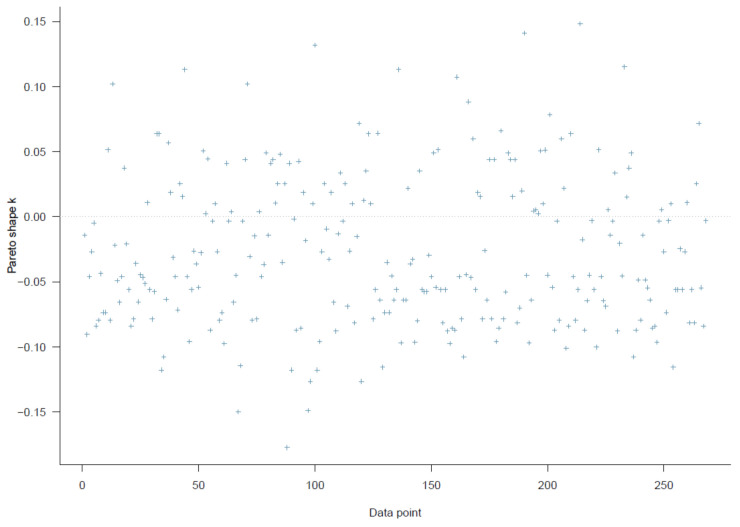
Model 4′s PSIS diagnostic plot.

**Figure 17 ijerph-18-03681-f017:**
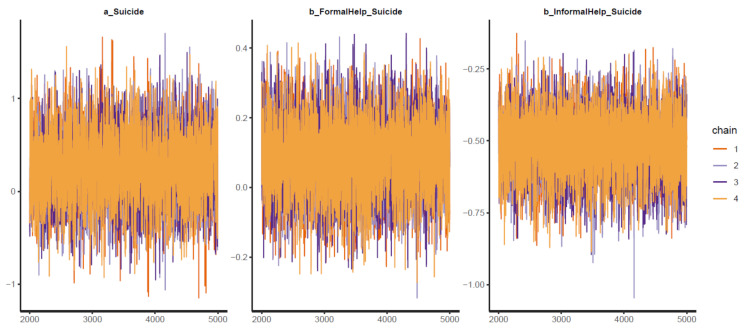
Trace plots for Model 4′s posterior coefficients.

**Figure 18 ijerph-18-03681-f018:**
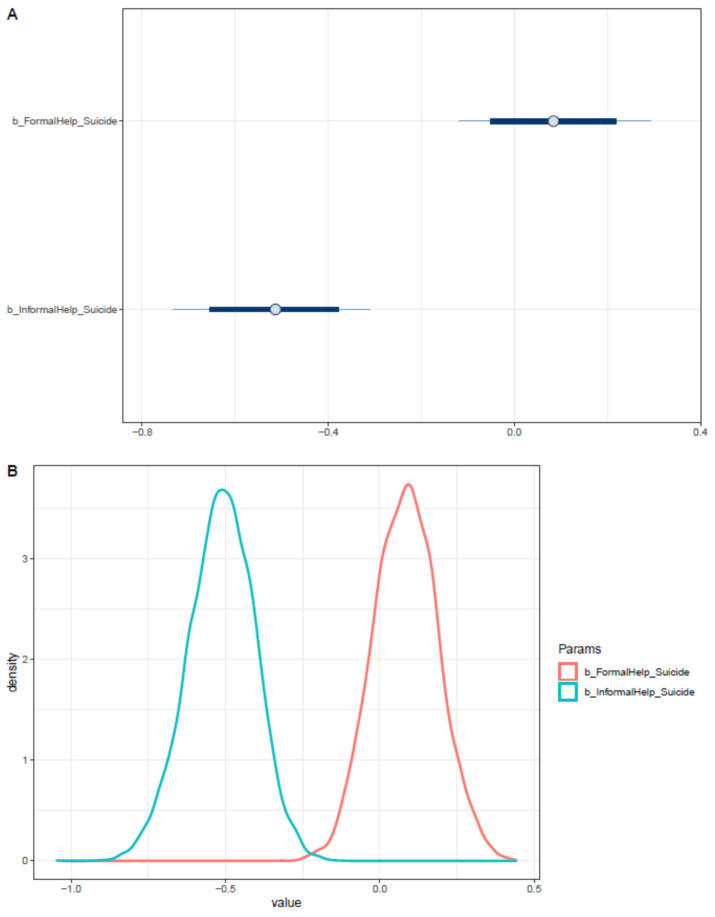
Distributions of Model 4′s posterior coefficients. (**A**)–Interval plot; (**B**)–Density plot.

**Figure 19 ijerph-18-03681-f019:**
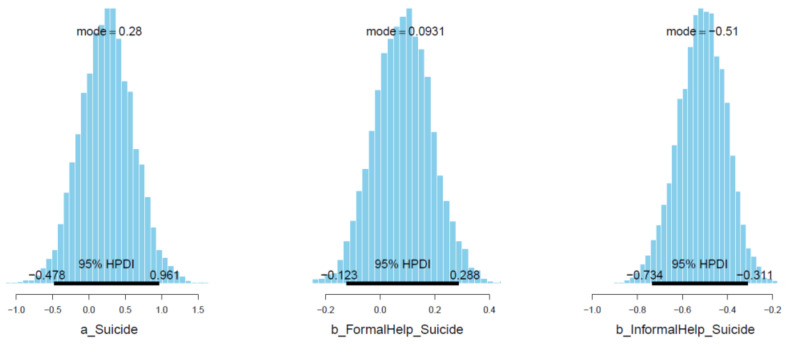
Distributions of Model 4′s posterior coefficients with HPDI at 95%.

**Table 1 ijerph-18-03681-t001:** Variable description.

Name	Variable	Data Type	Description
Suicidal ideation	*Suicide*	Binary	Suicidal ideation in the past two weeks based on the PHQ-9 [27,28]. ‘Yes’ is coded as 1, ‘No’ is coded as 0.
Informal help-seeking source	*InformalHelp*	Continuous	The average score of willingness to seek help from parents and family members. The data range from 1 to 7.
Formal help-seeking source	*FormalHelp*	Continuous	The average score of willingness to seek help from professionals and doctors when encountering emotional difficulties. The data range from 1 to 7.
Type of student	*Inter_Dom*	Binary	Whether the respondent is an international or domestic student. ‘International student’ is coded as 1, ‘Domestic student’ is coded as 0.
Total connectedness and companionship	*TCC*	Continuous	The total score of connectedness (4 items) and companionship (1 item) measured by the Social Connectedness Scale developed by Lee and Robins [29]. The data range from 6 to 48.

**Table 2 ijerph-18-03681-t002:** Model 1′s simulated posteriors.

Parameters	Mean	SD	n_eff	Rhat
*Constant*	2.56	0.65	4236	1
*TCC*	−0.93	0.16	3821	1
*TCC*Inter_Dom*	0.07	0.09	4932	1

**Table 3 ijerph-18-03681-t003:** Model 2′s simulated posteriors.

Parameters	Mean	SD	n_eff	Rhat
*Constant*	2.58	0.43	5593	1
*TCC*	0.31	0.10	5522	1
*TCC*Inter_Dom*	−0.09	0.04	7420	1

**Table 4 ijerph-18-03681-t004:** Model 3′s simulated posteriors.

Parameters	Mean	SD	n_eff	Rhat
*Constant*	3.32	0.42	5707	1
*TCC*	−0.06	0.09	5404	1
*TCC*Inter_dom*	−0.06	0.05	7202	1

**Table 5 ijerph-18-03681-t005:** Model 4′s simulated posteriors.

Parameters	Mean	SD	n_eff	Rhat
*Constant*	0.23	0.36	5916	1
*FormalHelp*	0.09	0.10	5809	1
*InformalHelp*	−0.51	0.10	5470	1

## Data Availability

The data that supports the findings of this study is available at https://www.mdpi.com/2306-5729/4/3/124/s1 (accessed on 1 January 2021).

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
