# Peer review of "Alice in Suicideland: Exploring the Suicidal Ideation Mechanism through the Sense of Connectedness and Help-Seeking Behaviors"

_ijerph, 2021, doi:10.3390/ijerph18073681_

Round 1
Reviewer 1 Report
The abstract should be a total of about 200. This abstract is more than 300 words.
Line 120 the dot is unnecessary (. Mindsponge mechanism)
Line 256: please, including the project identification code, date of approval from an ethics committee
Reviewer 2 Report
Comments to the Author.
The current study interest is, although the rate of suicide cases is high, many studies focused on the risk factors for suicidal behaviors, less attention given to the process of the transition from ideation to a suicide attempt.
Associations between Item 9 in the PHQ-9 (suicidal ideation) used with four variables: (i) suicidal ideation; (ii) help-seeking willingness; (iii) sense of connectedness; and (iv) information inaccessibility.
The finding shows that sense of connectedness is negatively associated with suicidal ideation, but when the interaction with international students is added, the sense of connectedness’s effect on suicidal ideation becomes less impactful; The effect of a sense of connectedness on help-seeking behavior toward family members among international students is also lessened; informal help-seeking (family members) is
negatively associated with suicide whereas formal help is positive. The findings support our assumption on three fundamental conditions for preventing suicidal thoughts: (i) a high degree of belongingness, (ii) accessibility to help-related information, and (iii) healthy perceived cultural responses towards mental health.
Abstract:
The overture of the abstract is very ambitious, I would present the purpose of the present study. Either I would point out that at the beginning of the suicidal process there are suicidal thoughts, or I would point out that suicidal thoughts are common among adolescents and are a risk factor for suicide attempt and suicide, then present the purpose of the present study.
Major weakness- I'm not sure the study points to a mechanism. The Bayesian analysis presented a cumulative association between risk factors (and interactions) to suicidal thoughts. If so, it should be presented this way.

Reviewer 3 Report
The publication is very original, unique and interesting.
Process connecting study concept of suicidal ideation with metaphor of reality perception by the author of "Alice in Wonderland" has an interesting level of abstraction.
The presented analysis of the mechanism of suicidal ideation requires further research. It is worth comparing the results obtained in various populations, including primarily those with diagnosed psychiatric disorders that are associated with the highest suicidal risk.
I only suggest adding a minor text correction.
What was the process of recruiting participants and their selection? How did the authors ensure that the students participating in the study were the target group to exhibit suicidal ideation? Did the questionnaire contain a history of mental disorders: depression, anxiety disorders, etc. If not, why was it abandoned? Please explain this algorithm.
Round 2
Reviewer 2 Report
Alice in Suicide land: Exploring the suicidal ideation mechanism through sense of connectedness and help-seeking behaviors
Minh-Hoang Nguyen, Tam-Tri Le, Hong-Kong To Nguyen, Manh-Toan Ho, Huyen T. Thanh Nguyen, and Quan-Hoang Vuong
Comments to the Author.
The current study interest is, although the rate of suicide cases is high, many studies focused on the risk factors for suicidal behaviors, less attention given to the process of the transition from ideation to a suicide attempt.
Associations between Item 9 in the PHQ-9 (suicidal ideation) used with four variables: (i) suicidal ideation; (ii) help-seeking willingness; (iii) sense of connectedness; and (iv) information inaccessibility.
The finding shows that sense of connectedness is negatively associated with suicidal ideation, but when the interaction with international students is added, the sense of connectedness’s effect on suicidal ideation becomes less impactful; The effect of a sense of connectedness on help-seeking behavior toward family members among international students is also lessened; informal help-seeking (family members) is
negatively associated with suicide whereas formal help is positive. The findings support our assumption on three fundamental conditions for preventing suicidal thoughts: (i) a high degree of belongingness, (ii) accessibility to help-related information, and (iii) healthy perceived cultural responses towards mental health.
Abstract:
The overture of the abstract is very ambitious, I would present the purpose of the present study. Either I would point out that at the beginning of the suicidal process there are suicidal thoughts, or I would point out that suicidal thoughts are common among adolescents and are a risk factor for suicide attempt and suicide, then present the purpose of the present study.
Major weakness- I'm not sure the study points to a mechanism. The Bayesian analysis presented a cumulative association between risk factors (and interactions) to suicidal thoughts. If so, it should be presented this way.
